# A Mobile Application to Perform the Six-Minute Walk Test (6MWT) at Home: A Random Walk in the Park Is as Accurate as a Standardized 6MWT

**DOI:** 10.3390/s22114277

**Published:** 2022-06-03

**Authors:** Martijn Scherrenberg, Cindel Bonneux, Deeman Yousif Mahmood, Dominique Hansen, Paul Dendale, Karin Coninx

**Affiliations:** 1Heart Centre Hasselt, Jessa Hospital, 3500 Hasselt, Belgium; martijn.scherrenberg@telenet.be (M.S.); dominique.hansen@uhasselt.be (D.H.); paul.dendale@jessazh.be (P.D.); 2Faculty of Medicine and Life Sciences, Hasselt University, 3590 Diepenbeek, Belgium; 3Faculty of Medicine, Antwerp University, 2000 Antwerp, Belgium; 4HCI and eHealth, Faculty of Sciences, Hasselt University, 3590 Diepenbeek, Belgium; cindel.bonneux@uniweb.be (C.B.); demo_y86@yahoo.com (D.Y.M.); 5BIOMED-REVAL-Rehabilitation Research Centre, Faculty of Rehabilitation Sciences, Hasselt University, 3590 Diepenbeek, Belgium

**Keywords:** six-minute walk test, 6MWT, six-minute walk distance, 6MWD, walking, cardiovascular disease, smartphone, digital health

## Abstract

The six-minute walk test (6MWT) provides an objective measurement of a person’s functional exercise capacity. In this study, we developed a smartphone application that allows cardiac patients to do a self-administered 6MWT at home on a random trajectory. In a prospective study with 102 cardiovascular disease patients, we aimed to identify the optimal circumstances to perform a smartphone-measured 6MWT, i.e., the best algorithm and the best position to wear the smartphone during the test. Furthermore, we investigated if a random walk is as accurate as a standardized 6MWT. When considering both the reliability and accuracy of the distance walked, the best circumstances to perform a standardized smartphone-measured 6MWT are wearing the smartphone in a strap around the patient’s arm and using an algorithm that relies on the processed step count data acquired from Google Fit. Furthermore, we demonstrated that a smartphone-measured walk along a random trajectory is as accurate to determine a cardiac patient’s functional exercise capacity as a standardized (smartphone-measured) 6MWT. We conclude this paper by presenting how our 6MWT application can be used in a home setting to remotely follow up on cardiac patients’ functional exercise capacity.

## 1. Introduction

The six-minute walk test (6MWT) is a frequently used test for the objective assessment of functional exercise capacity in patients [1]. In this safe, low-complexity test, patients are asked to walk as far as possible along a 30-m corridor for 6 min. The main endpoint is the six-minute walk distance (6MWD) [2]. Other endpoints are the modified Borg score and peripheral arterial oxygen saturation via pulse oximetry [1]. The 6MWT is often used as a low-cost and easy-to-perform alternative for maximal cardiopulmonary exercise testing (CPET) [3]. The main advantages of the 6MWT over the CPET to follow up on a patient’s functional exercise capacity are threefold: (1) a 6MWT is very easy to perform, (2) there is no need to buy expensive equipment to perform the test, and (3) a 6MWT can be performed outside a hospital environment and without the supervision of a healthcare professional, eliminating the need to come to the hospital to perform the test [1].

The strongest indication for the 6MWT is to measure the clinical evolution to medical and research interventions in patients with moderate to severe heart or lung disease [1]. The 6MWD is strongly associated with long-term cardiovascular outcomes [3,4]. The 6MWT is also an interesting follow-up tool of functional capacity in cardiac patients. As such, the 6MWT can measure progression or deterioration in physical fitness. A self-administered smartphone-based 6MWT could possibly facilitate regular testing of functional capacity in the patient’s home environment without supervision of a healthcare professional. A self-administered smartphone-based 6MWT could also be a useful follow-up tool in telerehabilitation and telemonitoring programs [5].

In the future, we envision the integration of a self-administered 6MWT in telemonitoring applications, supporting patients and healthcare professionals in remotely following up on the patient’s physical fitness. Since the start of COVID-19 pandemic, there has also been a growing demand for tools to remotely follow up on cardiac patients’ functional capacity. The following aims were investigated in this study.

Identify the best algorithm and position to wear the smartphone to perform a self-administered smartphone-measured 6MWT, by comparing different algorithms and positions to wear the smartphone during a standardized smartphone-measured 6MWT.Propose a simplified home-based 6MWT protocol.Assess cardiovascular disease patients’ opinion about using a smartphone application to perform the 6MWT at home.

The novelty of this study is threefold: (1) it is one of the first studies to investigate the accuracy of a smartphone-measured random walk 6MWT, (2) it is one of the first studies to research the impact of various smartphone positions on the accuracy of the smartphone-measured 6MWD, and (3) it is also one of the first studies to look into a smartphone-measured 6MWT in a group of cardiac rehabilitation patients.

## 2. Materials and Methods

### 2.1. Algorithms

Four different algorithms were programmed to measure the 6MWD. The programming was performed in Java and Kotlin (Android), while using the *Google Fit API* to acquire step count data. Two of the algorithms (I and II) were based on the patient’s measured step length and the step count determined with Google Fit. Both algorithms retrieve from Google Fit the person’s step count during the 6MWT and multiply this value with the person’s step length (that was entered in the 6MWT application prior to the start of the 6MWT). Algorithm I or *Google Fit Raw* uses the raw step count data collected by *Google Fit*, by directly reading by the data source. This means that the app only uses local step data, the step count will be different from the step count presented in the *Google Fit* app. Algorithm II or *Google Fit Processed* uses the step count data collected and processed by *Google Fit*, by reading by data type. This means that the *Google Fit* platform reviews all the information available for a specific data type from different sources and merges it in a logical way. According to *Google Fit* [6], reading by data type provides a more accurate estimated step count than using the raw *Google Fit* data.

Algorithms III and IV use the smartphone’s Global Positioning System (*GPS*) to calculate the distance walked. Both algorithms are based on the outdoor distance estimation algorithm of Salvi et al. [7]. During the 6MWT, every 500 milliseconds, the smartphone’s position is requested from the built-in *GPS* sensors. In Algorithm III or *GPS*, the distance between the current location and the previous location is added to the current distance walked. As such, the total 6MWD is accumulated at the end of the 6MWT. Algorithm IV or *GPS and Google Fit* uses the step count data from *Google Fit* along with the *GPS* data to make a more accurate distance estimation than only relying on the built-in *GPS* sensors. If the number of steps did not change between two *GPS* measurements, the distance walked is not counted.

### 2.2. MWT Application

Before starting the 6MWT, the application encapsulating the algorithms asks the healthcare professional to enter the patient’s step length (i.e., needed for algorithms I and II). When the button to start the 6MWT is tapped, audio feedback is provided and a countdown in seconds from 10 to 0 (i.e., the start of the test) is initiated. During the 6MWT, audio feedback is provided when the patient is halfway (i.e., after 3 min) and when the test ends. After the 6MWT has ended, the 6MWDs of the four different algorithms are shown on the screen and saved on the Samsung smartphone.

### 2.3. Study Design

The study is a prospective study with 102 cardiovascular disease patients with an indication for cardiac rehabilitation. Patients are recruited in Jessa Hospital in Belgium. The study was approved by the medical ethical committees of the Jessa Hospital and the University of Hasselt.

#### 2.3.1. Participants

All cardiovascular disease patients that are enrolled in phase II center-based cardiac rehabilitation are eligible for participation in the study. Phase II cardiac rehabilitation is defined as the center-based intervention for secondary prevention performed following the index cardiovascular disease event with the aim of clinical stabilization, risk stratification, and promotion of a long-term health status [8].

#### 2.3.2. Study Procedure

As part of the rehabilitation session, the patient was asked to participate in the study. When the patient was willing to participate in the study, the informed consent form was signed. During the study, the participant was guided by a healthcare professional (e.g., trainee physiotherapy, physiotherapist, or cardiologist in training) who was familiar with the study procedure. Once the participant gave consent to be included in the study, the study procedure consisted of the following steps:In the rehabilitation center, basic demographic data about the patient (i.e., name, date of birth, gender, height) were collected and written down on a results sheet.Outside, in the park near the rehabilitation center, the patient’s step length was determined by counting how many steps the patient did when walking 30 m straight on a predefined path. The healthcare professional divided 30 m (i.e., the distance walked) by the number of steps taken by the patient to calculate the patient’s step length. The step length was rounded to the nearest centimeter. The healthcare professional noted the patient’s step length on the results sheet.The patient performed the standardized 6MWT [1] in the park, on a fixed, straight path of 30 m. The smartphone application was started by the healthcare professional. The distance walked was measured by a healthcare professional by counting how many times the patient walks along the fixed 30-m path. For the last length of the 6MWT, the distance walked was measured manually by the healthcare professional. The measured 6MWD was noted on the results sheet. At the same time, the smartphone measured the 6MWD.The patient had a short break. It was up to the patient to decide when he/she felt ready to perform the second 6MWT.The patient performed a random walk 6MWT in the park. The patient was instructed to walk freely, wherever he/she wanted (starting from the location in the park where the standardized 6MWT was performed), but still to walk as far as possible for 6 min.Back at the rehabilitation center, the patient filled in a short custom-made questionnaire on paper. The questionnaire consisted of several sections assessing the patient’s acceptance (e.g., patient’s willingness to do a 6MWT at home) and preferred conditions to do a 6MWT in their home environment (e.g., position to wear the smartphone and walking trajectory).

During the 6MWTs, the patient wore three smartphones (Figure 1): smartphone A was worn in a case mounted on the patient’s arm, smartphone B was held in the patient’s hand, and smartphone C was worn in the front pocket of the patient’s pants. If patients did not have a front pocket in their pants, the smartphone was put in a belt that is worn around the waist. The healthcare professional supported the patient in putting the smartphones in the different positions. Furthermore, the healthcare professional had a master smartphone that could trigger the start of the 6MWT on the different smartphones simultaneously via Bluetooth. As such, the healthcare professional could trigger the start of the 6MWT after all smartphones were correctly positioned on the patient’s body.

### 2.4. Selection of the Best Algorithm and Smartphone Position

Several aspects should be considered to determine the best algorithm–smartphone position combination for a smartphone-measured self-administered 6MWT. A first important aspect to consider when deciding upon the best combination is the accuracy. We use the distance that was measured manually by the healthcare professional in a standardized manner [1] as the reference distance for both the smartphone-measured standardized 6MWT and the smartphone-measured random walk 6MWT. Different measures related to the accuracy and error rate can be used, such as the mean absolute error (MAE), the error rate, the median, and the standard deviation (SD) of the absolute error.

Secondly, the reliability of the algorithm is important. Therefore, we should investigate the circumstances in which the algorithm provides no result (missing value) or an invalid result (outlier). Different methods can be considered to detect outliers, such as a fixed threshold that is the same for all participants, or a relative amount of error, calculated as a percentage of the patient’s reference distance walked (i.e., if the result is X% different from the healthcare professional-measured distance walked). In our analyses, we chose to use the following data set: personalized-threshold, outlier-filtered data set: an outlier-filtered data set, with all 6MWDs smaller than 50% of the patient’s healthcare professional-measured 6MWD considered as an outlier.

The rationale for this decision is as follows: outlier detection based on a personalized threshold is the most realistic, because it is dependent on the actual distance walked during the 6MWT by the individual.

For each trajectory (i.e., standardized and random walk), we calculated for each algorithm–smartphone position combination the following metrics:Mean absolute error (MAE), i.e., the mean of the absolute difference between the 6MWDs as measured by the healthcare professional, and as measured by the algorithm on the smartphone.Median error, i.e., the median of the difference between the 6MWDs as measured by the healthcare professional, and as measured by the algorithm on the smartphone.Standard deviation (SD) of the absolute error, i.e., the standard deviation of the absolute difference between the 6MWDs as measured by the healthcare professional, and as measured by the algorithm on the smartphone.Error rate, i.e., the proportion of 6MWDs that are incorrectly measured.

Minimal clinically important difference (MCID) is the smallest difference in score in the domain of interest which patients perceive as beneficial and which would mandate, in the absence of troublesome side effects and excessive cost, a change in the patient’s management. The review of Bohannon and Crouch [9] indicates that the MCID of the 6MWT is dependent on pathology, and thus in our study dependent on the type of cardiovascular disease. Bohannon and Crouch [9] suggest that changes in six-minute walk distance exceeding 30.5 m can be considered clinically meaningful.

## 3. Results

A total of 103 cardiovascular disease patients were recruited. One patient was excluded from the data analysis, due to stopping the 6MWT prematurely for a medical reason (cramps in his/her legs). All patients that were included in the data analysis performed both the smartphone-measured, standardized 6MWT and the smartphone-measured, random walk 6MWT. Participants’ age ranged from 27 to 90 years (mean age = 65.9 years, standard deviation = 11.4 years) with 23 female and 79 male participants. Included patients were participating in a cardiac rehabilitation program due to recent coronary artery disease, heart failure, recent ablation, recent pacemaker, or a cardiac implantable electronic device implantation.

### 3.1. Smartphone-Measured, Standardized 6MWT

The results of 102 standardized 6MWTs running the four algorithms in parallel (*Google Fit Raw, Google Fit Processed, GPS, GPS and Google Fit*) in the three smartphone positions (*pocket, hand, arm*) were used to determine the accuracy and reliability of the smartphone-measured, standardized 6MWT. Table 1 depicts the analysis of the personalized-threshold, outlier-filtered data set.

As depicted in Table 1, the algorithm *Google Fit Processed* with the smartphone mounted on the patient’s *arm* yielded for the personalized-threshold, outlier-filtered data set the smallest mean absolute error, the best median error, the minimum standard deviation, and the minimum error rate. Therefore, we can conclude that among the algorithms that we included in this study, the algorithm *Google Fit Processed* with the smartphone mounted on the patient’s *arm* is the best combination for a smartphone-measured, standardized six-minute walk test.

The mean absolute error for the algorithm *Google Fit Processed* with the smartphone worn in a strap around the *arm* is 20.16 m, which is lower than the MCID suggested by Bohannon and Crouch [9]. Furthermore, the maximal absolute error for this algorithm–smartphone position combination is 66 m. This indicates that this algorithm–smartphone position combination can be used to detect changes in cardiovascular disease patients’ functional capacity.

In Figure 2, the 6MWDs, as measured by the smartphone with the *Google Fit Processed*–*arm* combination, and as measured by the healthcare professional manually, are depicted for each participant. For the majority of the patients, the 6MWD measured by the healthcare professional and the 6MWD measured by the algorithm *Google Fit Processed* in the *arm* position are overlapping or extremely close to each other, indicating that the smartphone-measured 6MWD is fairly accurate. For this algorithm–smartphone position combination, the number of outliers is limited (only two outliers). Missing values for this algorithm–smartphone position combination are indicated by no value/dot for the participant (e.g., participant 5). Possible reasons for missing values include the smartphone going into sleep mode during the 6MWT and the patient pressing a button in the 6MWT application while the test is ongoing.

The Bland–Altman plot (Figure 3) shows the difference between the 6MWD measured manually by the healthcare professional, and the 6MWD measured by the algorithm *Google Fit Processed* in the *arm* position. The difference in meters between the healthcare professional-measured and smartphone-measured 6MWD for this algorithm–smartphone position combination is at most 66 m. Relative to the total distance walked during the 6MWT, which is minimally 228 m and maximally 657 m, this difference is acceptable. Furthermore, for most participants, the difference is significantly smaller.

Next, using the same metrics (i.e., minimum mean absolute error, best median, minimum standard deviation, and minimum error rate), we analyzed which algorithm performs best in each position for a smartphone-measured, standardized 6MWT. The algorithm *Google Fit Processed* outperformed all other algorithms in all positions (i.e., *pocket, hand, arm*) by recording the minimum mean absolute error, the best median, the minimum standard deviation, and the minimum error rate (Table 1), indicating that the best algorithm to use for a smartphone-measured, standardized 6MWT is *Google Fit Processed*, independent of the position of the smartphone.

Similarly, we inspected our evaluation metrics (i.e., minimum mean absolute error, best median, minimum standard deviation, and minimum error rate) for each algorithm separately in all positions to find the best position to wear the smartphone for each algorithm for a smartphone-measured, standardized six-minute walk test (Table 1).

The algorithm *Google Fit Raw* obtained the best results with the smartphone in the front *pocket* of the patient’s pants. Since the accuracy of this algorithm heavily depends on an accurate detection of steps, it is logical that a position that allows accurate step detection (such as the pocket of your pants) yields the best results. The algorithm *Google Fit Processed* produced the most accurate results when the smartphone was mounted in a strap around the *arm*. This is surprising, since similar to the algorithm *Google Fit Raw*, the accuracy of this algorithm heavily depends on the accurate detection of steps. However, the algorithm *Google Fit Processed* uses Google Fit’s preprocessing of steps, which gives a more accurate step count than using the raw *Google Fit* step count data [10] (as in the *algorithm Google Fit Raw*). This might have an influence on the recognition of steps in the different smartphone positions and thus the best position to wear the smartphone for the two algorithms that are based on *Google Fit* data.

The algorithm *GPS* achieved the best results with the smartphone in the *hand*. We think this could be caused by the localization data being more accurate when the smartphone is held in the hand (no interference from other material surrounding the smartphone, such as the arm strap or the patient’s pants), in comparison with the smartphone worn in a strap around the arm or in the pocket of the patient’s pants. The algorithm *GPS and Google Fit* performed best with the smartphone carried in the *hand*. It is reasonable to expect that the best results for both the algorithms *GPS* and *GPS and Google Fit* are the same, since both rely mostly on the localization information provided by the smartphone’s built-in *GPS* sensors to measure the distance walked during the six-minute walk test.

In addition to the accuracy of the measured six-minute walk distance, it is important to consider the reliability of the algorithm–smartphone position combination to evaluate which combination is best for a smartphone-measured, standardized 6MWT. Table 2 provides for each algorithm–smartphone position combination an overview of the number of missing values (i.e., the algorithm provides no result for the 6MWD) and the number of outliers (i.e., the algorithm provides an unrealistic 6MWD). It is remarkable that there are no outliers detected for the algorithm *Google Fit Raw*. Most outliers were detected for the combination *GPS and Google Fit–pocket*, indicating that this is the least reliable algorithm–smartphone position combination in terms of outliers.

When considering the algorithm that performed best in each smartphone position, i.e., *Google Fit Processed*, we can conclude that for this algorithm the number of outliers is quite limited, ranging from zero to two outliers dependent on the smartphone position. In terms of reliability, the best algorithm–smartphone position combinations are the algorithm *Google Fit Raw* in the *hand* position and the algorithm *GPS* in the *hand* position. For all algorithms, the number of missing values and number of outliers are lowest when the smartphone is carried in the *hand*. However, it must be noted that the difference in the number of missing values and outliers is small for most algorithm–smartphone position combinations, with the exception of the combination *GPS and Google Fit–pocket*.

### 3.2. Smartphone-Measured, Random Walk 6MWT

The results of 102 random walk six-minute walk tests (6MWTs) running in parallel the four algorithms (*Google Fit Raw, Google Fit Processed, GPS, GPS and Google Fit*) in the three smartphone positions (*pocket, hand, arm*) were used to determine the best algorithm–smartphone position combination for a smartphone-measured, random walk 6MWT. We considered both the reliability and the accuracy of the six-minute walk distance (6MWD) to determine the best algorithm–smartphone position combination. We used the 6MWD as measured by the healthcare professional during the standardized 6MWT as a reference to determine the accuracy. Table 3 shows the analysis of the 6MWD for a random walk six-minute walk test for the personalized-threshold, outlier-filtered data set.

The algorithm *Google Fit Processed* with the smartphone worn in a strap around the patient’s *arm* produced the smallest mean absolute error, the best median error, the minimum standard deviation, and the minimum error rate. As a result, we can conclude that among the algorithms that we included in this study, the algorithm *Google Fit Processed* with the smartphone mounted on the patient’s *arm* provides the most accurate 6MWD for a smartphone-measured, random walk six-minute walk test.

When comparing the mean absolute error for the algorithm *Google Fit Processed* with the smartphone worn in a strap around the patient’s *arm* (i.e., 20.6 m), with the minimal clinically important difference (i.e., 30.5 meters [9]), we can conclude that this algorithm–smartphone position combination can be used to detect changes in cardiovascular disease patients’ functional capacity. Furthermore, the maximal absolute error for this algorithm–smartphone combination is 111.2 m.

In Figure 4, the six-minute walk distances (6MWDs), as measured during the random walk 6MWT by the smartphone with the algorithm *Google Fit Processed* with the smartphone mounted in a strap around the patient’s *arm*, and as measured during the standardized 6MWT manually by the healthcare professional, are depicted for each participant. For most participants, the 6MWD measured by the healthcare professional during the standardized 6MWT and the 6MWD measured by the algorithm *Google Fit Processed* in the *arm* position during the random walk 6MWT are extremely close to each other or even overlapping, indicating that the smartphone-measured 6MWD is fairly accurate. For this algorithm–smartphone position combination, there are only two outliers. Missing values for this combination of algorithm–smartphone position are indicated by no dot for the participant (e.g., participant 5).

Figure 5 depicts the Bland–Altman plot, showing the difference between the 6MWD measured by the algorithm *Google Fit Processed* in the *arm* position during the random walk 6MWT, and the 6MWD measured manually by the healthcare professional during the standardized 6MWT. The difference in meters between the healthcare professional-measured, standardized 6MWD and smartphone-measured, random walk 6MWD for this algorithm–smartphone position combination is at most 111.2 m. Relative to the total distance walked during the random walk 6MWT, which is minimally 262.1 m and maximally 635 m, this difference is acceptable. Furthermore, for most participants, the difference is significantly smaller, indicated by the dots in the Bland–Altman plot (Figure 5) being close to 0.

Using the same metrics (i.e., minimum mean absolute error, best median, minimum standard deviation, and minimum error rate), we determined which algorithm performs best in each position for a smartphone-measured, random walk 6MWT. The algorithm *Google Fit Processed* surpassed all other algorithms in all three smartphone positions (*pocket, hand, arm*) by recording the minimum mean absolute error, the best median, the minimum standard deviation, and the minimum error rate (Table 3).

Likewise, we inspected our evaluation metrics (i.e., minimum mean absolute error, best median, minimum standard deviation, and minimum error rate) for each algorithm separately in all positions to find the best position to wear the smartphone for each algorithm for a smartphone-measured, random walk 6MWT (Table 3).

The algorithm *Google Fit Raw* obtained the best results with the smartphone in the front *pocket* of the patient’s pants, whereas the algorithm *Google Fit Processed* performed best with the smartphone mounted in a strap around the patient’s *arm*. The result for the algorithm *Google Fit Processed* is surprising, since we expected that a position in which steps can be accurately detected is favorable for these two algorithms that heavily depend on accurate step count detection (i.e., *Google Fit Raw* and *Google Fit Processed*).

The algorithms *GPS* and *GPS and Google Fit* produced the best results with the smartphone in the *pocket* position. As expected, the position to wear the smartphone for both algorithms that depend on the localization information provided by the built-in *GPS* sensors (i.e., *GPS* and *GPS and Google Fit*) is the same. However, we do not have a direct indication of why the *pocket* position performs better than the other two smartphone positions.

Similar to the smartphone-measured, standardized 6MWT, we investigated the reliability of the different algorithm–smartphone position combinations to evaluate which combination is best for a smartphone-measured, random walk 6MWT. Table 4 provides for each investigated algorithm–smartphone position combination an overview of the number of missing values and the number of outliers. It is noteworthy that no outliers were detected for the algorithm *Google Fit Raw* in any of the smartphone positions. Furthermore, several other algorithm–smartphone position combinations did not have any outliers: *Google Fit Processed–pocket*, *GPS–pocket*, *GPS–arm*, and *GPS and Google Fit–arm*). Most outliers were detected for the algorithm *GPS and Google Fit* with the smartphone worn in the front *pocket* of the patient’s pants, indicating that this is the least reliable algorithm–smartphone position combination in terms of outliers.

When considering the algorithm that performed best in each smartphone position, i.e., *Google Fit Processed* (Table 3), we can conclude that for this algorithm the number of outliers is quite low, ranging from zero (for the *pocket* position) to two outliers (for the *arm* position) dependent on the smartphone position. In terms of reliability, the best algorithm–smartphone position combinations are *GPS–arm* and *GPS and Google Fit–arm*. For all algorithms, the total number of missing values and outliers is lowest when the smartphone is carried in the *arm* position and highest when the smartphone is carried in the front *pocket* of the patient’s pants. A possible explanation is that missing values might be caused by accidentally pressing a button while the 6MWT is ongoing, which happens easily when the smartphone is carried in the pocket of the patient’s pants.

### 3.3. Comparison of the Smartphone-Measured Standardized 6MWT and the Smartphone-Measured Random Walk 6MWT

For both trajectories, the most accurate algorithm–smartphone combination for a smartphone-measured 6MWT is the algorithm *Google Fit Processed* with the smartphone mounted in a strap around the patient’s *arm*. As a result, this is the most accurate algorithm–smartphone position combination, independent of the walking trajectory. In addition, our analysis demonstrated that for a smartphone-measured 6MWT the algorithm *Google Fit Processed* outperforms all other algorithms for all smartphone positions, independent of the walking trajectory.

Figure 6 presents the six-minute walk distance (6MWD) as measured by the algorithm *Google Fit Processed* in the *arm* position for both the standardized 6MWT and the random walk 6MWT. In addition, the reference distance, i.e., the 6MWD as measured manually by the healthcare professional during the standardized 6MWT, is depicted. For most participants, both smartphone-measured 6MWDs are very close to or even overlapping with the healthcare professional-measured 6MWD. With respect to the outliers, we can see that for participant 56 the smartphone-measured 6MWD is considered an outlier for both walking trajectories, whereas the second outlier is unique for each walking trajectory (i.e., 6MWD value of participant 13 for the standardized 6MWT and 6MWD value of participant 81 for the random walk 6MWT).

The smartphone-measured, standardized 6MWT is slightly more accurate than the smartphone-measured, random walk 6MWT, indicated by a lower mean absolute error, a lower standard deviation of the absolute error, and a lower error rate. However, when considering this with respect to the minimal clinically important difference, we can conclude that this difference in accuracy is not significant. We see that the smartphone-measured random walk 6MWT rather underestimates the 6MWD, whereas the smartphone-measured standardized 6MWT rather overestimates the 6MWD. As already mentioned, the 6MWD gives an indication of a patient’s functional capacity. Therefore, it can be used as an information source when prescribing physical exercise training to cardiac patients. For physical exercise training, it is safer to underestimate a person’s functional capacity and thus give a lower training, rather than overestimate a person’s functional capacity and thus prescribe a too intense exercise program. However, it must be noted that the difference is only very small.

For both walking trajectories (standardized and random walk), the reliability turns out to be similar when considering the most accurate algorithm–smartphone position combination. Indeed, the number of outliers is exactly the same for both walking trajectories. On the contrary, the number of missing values is higher for the smartphone-measured standardized 6MWT (15 missing values) than for the smartphone-measured random walk 6MWT (four missing values). However, it is unclear if these outliers are due to the walking trajectory, since missing values were often caused by patients accidentally pressing a button on the smartphone while the 6MWT was ongoing. Furthermore, the best algorithm–smartphone position combinations in terms of reliability are different for the two walking trajectories.

### 3.4. Patients’ Opinion about a Self-Administered 6MWT

In our study, the 6MWT application was controlled by the healthcare professional. Patients’ opinion about performing a self-administered, smartphone-measured 6MWT alone at home was assessed with a survey. After the participants performed both the standardized 6MWT and the random walk 6MWT, they filled in a custom-made questionnaire consisting of several Likert scale, multiple choice, rating, and open questions. The questionnaire was completed by all 102 patients that performed the two 6MWTs. It is important to note that 10% of the participants clearly indicated that they are not using a smartphone regularly.

First, we assessed patients’ impressions and feelings about the 6MWT in general by asking them to indicate their opinion for a number of statements on a 5-point Likert scale, ranging from strongly disagree to strongly agree (Figure 7). The majority of the patients do not consider the 6MWT a tool that is only useful for healthcare professionals. The 6MWT is rather perceived as a tool that triggers reflection on their physical fitness, and thus could contribute to the motivation of the patient. Only three patients indicated that they do not like to perform the 6MWT to follow up on their progress themselves. Interpreting the result of their 6MWT is a challenge for some patients, indicated by a higher number of (strongly) disagree and neutral responses. This highlights the importance of education and guidance when patients perform the 6MWT in their home environment, without the supervision of a healthcare professional.

Next, we assessed patients’ opinion about using a smartphone application to perform a self-administered 6MWT at home (e.g., do they find it reliable to use a smartphone app to perform the 6MWT). Again, we used 5-point Likert scale questions, ranging from strongly disagree to strongly agree (Figure 8). It is notable that only five out of the 102 participants did not want to use a smartphone application to perform a 6MWT. Most patients think it is reliable and are not afraid to perform a self-administered, smartphone-based 6MWT without the supervision of a healthcare professional. We observed no clear difference in the moment (during or after finishing the rehabilitation program in the rehabilitation center) at which patients would find it more useful to perform a home-based 6MWT to follow up on their physical fitness. Patients would like that their healthcare professionals (e.g., cardiologist or physiotherapist) follow up the results of the 6MWTs that they perform at home. Lastly, we want to note that the negative and neutral responses regarding the usage of a smartphone application to do a 6MWT might be due to patients’ unfamiliarity with technology and smartphones in particular, or their unwillingness to follow up their physical fitness using a smartphone application.

As a next step, we investigated patients’ preferences for the location to perform a 6MWT, i.e., in the hospital/rehabilitation center under the supervision of a healthcare professional or at home using a smartphone application. Twenty-seven percent of the patients preferred to do the 6MWT in the rehabilitation center under the supervision of a healthcare professional, whereas 29 percent of the patients preferred to do the 6MWT at home with the smartphone application. Eighteen percent of the patients wanted to perform the 6MWT in both settings (in the rehabilitation center and at home). The remaining 25 percent did not have a preference for the location to perform the 6MWT. This indicates that only 27 percent of the participants do not want to perform a self-administered, smartphone-based 6MWT at home. We believe that this is due to the patients’ unfamiliarity with smartphones.

Currently, in the rehabilitation center, patients perform maximal cardiopulmonary exercise testing (CPET) three times during their rehabilitation trajectory. However, as mentioned in the Introduction, we investigated if the 6MWT could be used as an alternative to the CPET to follow up on the patient’s functional exercise capacity. In the questionnaire, we assessed patients’ preferences for the CPET versus the 6MWT. Out of the 98 participants that answered this question, 46 participants (i.e., 47%) preferred the 6MWT, 39 participants (i.e., 40%) preferred the CPET, and 13 participants (i.e., 13%) did not have a preference. Reasons for preferring the 6MWT included the easiness and pleasantness of the test. Reasons for preferring the CPET included preferring cycling over walking and the greater amount information that is available from the CPET.

As a main contribution of this paper, we investigated the influence of the walking trajectory (i.e., standardized or random walk) on the 6MWT results. To complement these analyses, we assessed patients’ preferences for the walking trajectory. The majority of the patients (66 out of the 102 participants, 65%) preferred the random walk 6MWT, whereas only 16 patients (16%) preferred the standardized 6MWT. Fourteen patients (14%) thought both walking trajectories were equally convenient. From these findings, we can conclude that most participants prefer the random walk trajectory over the standardized trajectory for a 6MWT.

To gather insight into patients’ preferences for the position to wear the smartphone (*pocket*, *hand*, *arm*), we included in the questionnaire some questions that asked the participants to rate the different options. For both walking trajectories, the results were quite similar. Most patients preferred to wear the smartphone in a strap around the *arm* (59 participants for the standardized 6MWT and 54 participants for the random walk 6MWT), followed by the front *pocket* of the patients’ pants, and lastly the *hand* position. Only a few patients did not have any preference. Some reasons for participants’ preferences for the *arm* position were that this was the least disturbing, distracting, or hindering position, there was less chance of dropping the smartphone, they still had freedom to move easily, and they did not have to carry the device in their *hand.* Arguments against the *arm* position included needing an extra accessory (i.e., *arm* strap) and that the *arm* strap could be inconvenient when wearing a pullover or a jacket.

### 3.5. Patients’ Opinion about a 6MWT Application Design

Given the study design and the aim of this study (i.e., finding the best algorithm–smartphone position combination to perform a smartphone-measured 6MWT), our developed 6MWT application was not designed to be used by cardiovascular disease patients alone in a home setting. When transforming our 6MWT application into an application enabling CVD patients to perform a self-administered 6MWT at home, some updates should be made (e.g., only one algorithm should be included and the UI should be optimized for usage by CVD patients). To gain insight into which additional features CVD patients would find useful in a 6MWT application, we presented to the participants some mockups of such an application (Figure 9). We assessed patients’ opinion regarding feedback during the 6MWT, progress follow-up after the 6MWT and walking speed details after the 6MWT. As indicated in Figure 10, most participants were quite positive about the mockups of the 6MWT application and would find it useful if a 6MWT application would offer these features (e.g., overview of the 6MWT progress and speed during the 6MWT).

## 4. Discussion

### 4.1. Summary of Study Results

In this study, we investigated different algorithms (*Google Fit Raw, Google Fit Processed, GPS, GPS and Google Fit*), positions to wear the smartphone (*pocket, hand, arm*), and walking trajectories (standardized and random walk) for a smartphone-measured 6MWT. When considering both the accuracy and the reliability, the algorithm *Google Fit Processed* in the *arm* position performed best, independent of the walking trajectory. Moreover, the algorithm *Google Fit Processed* outperformed all other algorithms in all three positions (*pocket, hand, arm*) for both the standardized and the random walk 6MWT. Therefore, we suggest to use the algorithm *Google Fit Processed* when developing an Android application to perform a smartphone-measured 6MWT. In addition, we advise patients to wear the smartphone in a strap around their *arm* to achieve the most accurate results. This is in line with the questionnaire results that indicated that patients preferred the *arm* position over the *pocket* and *hand* positions. A potential advantage of using an algorithm based on step count instead of *GPS* is that the test potentially also could be used indoors (e.g., shopping mall). This could be an advantage in many warmer countries where it is too hot to exercise during a significant proportion of the year.

The MAE was only 20.6 m for the *Google Fit Processed algorithm* with the smartphone worn in a strap around the patient’s *arm*. This MAE in this study is comparable to several other studies performing remote 6MWTs. The study of De Canniere et al. [10] demonstrated an MAE of 42.8 m using an accelerometer device, Salvi et al. demonstrated an MAE of 14 m using a smartphone 6MWT application [7]. Moreover, the MAE of only 20.6 m is lower than minimal clinical difference of 30.5 m [9].

Therefore, the main contribution of this article is that a random walk smartphone-measured 6MWT is as accurate as a standardized 6MWT supervised by a healthcare professional. This provides opportunities to ease the protocol for the 6MWT in a home setting. Using an updated version of our 6MWT application (e.g., inclusion of only one algorithm and addition of patient-facing features), cardiovascular patients’ functional exercise capacity can be assessed in an easy and accessible way, by only using a smartphone. The rigid requirement of walking along a 30-m corridor can be loosened to walking along a trajectory that the patient finds comfortable. As such, the patient can perform the 6MWT at any location where there is enough space to walk freely, e.g., in a park, on the sidewalk next to the road, or even in his/her garden. However, when choosing a location to perform the 6MWT, we advise that patients select a location where they are not hindered during the performance of the test, e.g., avoiding obstacles and crowded locations. In our questionnaire, we also assessed cardiovascular disease patients’ preferences for the walking trajectory (standardized or random walk). The majority of the patients preferred the random walk 6MWT, highlighting patients’ preference to have no restriction on the walking trajectory.

During the study, we observed quite some variation in the walking routes of the participants when performing the random walk 6MWT. Some participants walked completely random through the park, whereas others walked in circles, or just back and forth on a (more or less) fixed path. The walking route of a participant might have an influence on the accuracy and reliability of the algorithm. This is something we plan to investigate further in the future.

Giving patients freedom in where to perform the 6MWT makes it easier for them to perform the test outside the hospital or rehabilitation center. Furthermore, a free or random walk test could feel more natural to participants and thereby increase the odds that patients will perform it regularly.

As indicated by patients’ desire to perform the 6MWT to follow up on their progress themselves, we expect that patients will perform the smartphone-measured 6MWT regularly. The result of the 6MWT (i.e., 6MWD) can be used by healthcare professionals to determine the patient’s functional exercise capacity. As a consequence of the more frequent measurements, caregivers can monitor the patient’s functional exercise capacity more closely and timely adapt the exercise training to the patient’s capabilities. Updating the exercise training in a timely manner can in the end lead to better outcomes for the patients, e.g., faster recovery from their cardiovascular event. Furthermore, patients expressed that performing the 6MWT encourages them to think about and reflect on their physical fitness. Moreover, the results of their 6MWT motivate them to exercise.

### 4.2. Limitations

One of the limitations of our study is that the 6MWD during the random walk 6MWT was not measured manually by the healthcare professional. Therefore, we compared the smartphone-measured, random walk 6MWD with the 6MWD that was measured manually by the healthcare professional during the standardized 6MWT. However, since the goal is to have a valid alternative for a standardized 6MWT, it is legitimate to compare the smartphone-measured, random walk 6MWD with the healthcare professional-measured 6MWD of the standardized 6MWT.

Since the 6MWTs were performed in a park, there were other people walking by during the execution of the test. This might have hindered or distracted some of the participants while performing the 6MWT. However, this influences both the 6MWD as measured manually by the healthcare professional as well as the smartphone-measured 6MWD. Therefore, we expect this to have only a minor influence on the results. Patients who lacked pockets in their pants were instructed to wear the smartphones on a belt around their waist. It is possible that this influenced the results because movement in the pocket differs from movement around the waist.

Lastly, when testing the application during the study, we noticed that sometimes the 6MWT application stopped working when the smartphone was locked (accidentally) and/or the application was moved to the background. Since this affected the number of missing values and thus also the reliability of the algorithms, this is an issue that should be fixed before our 6MWT application can be used by patients in a home setting.

### 4.3. Future Directions

We envision that in the future a smartphone-measured 6MWT will be integrated into telemonitoring applications, supporting patients and healthcare professionals in remotely following up on the patient’s physical fitness [11,12]. In this subsection, we highlight some requirements for integrating a smartphone-measured 6MWT in telemonitoring applications.

Our current 6WMT application requires that, preceding the 6MWT, the patient’s step length is registered. Determining their step length by a health professional might be cumbersome for patients, i.e., setting out a path of 30 m and performing the calculations. A possibility to facilitate this is to incorporate in the application a formula that automatically calculates the patient’s step length based on some patient parameters. Equation (1) depicts a formula that can be used to estimate a person’s step length based on his/her gender and height (in centimeters) [13].
*female: stepLength = round(height ***** 0.413),*(1)
*male: stepLength = round(height ***** 0.415).*

As indicated in the study design, the healthcare professional determined the participant’s step length by counting how many steps the patient does when walking 30 m straight on a predefined path. The healthcare professional divided 30 m (i.e., the distance walked) by the number of steps taken by the patient to calculate the patient’s step length. The step length was rounded to the nearest centimeter. In addition, we collected study participants’ gender and height. Based on these data, we calculated the patient’s step length using Equation (1). To analyze how accurate this formula is to calculate the patient’s step length, we used the step length that was manually measured by the healthcare professional as the reference step length.

When considering these results in relation to the manually measured minimal and maximal step length, i.e., 48 cm and 91 cm, respectively, we can consider the mean absolute error of 5.24 cm and error rate of 0.07 quite reasonable. However, the algorithm *Google Fit Processed* heavily depends on an accurate step length to be able to calculate an accurate 6MWD. Therefore, a trade-off has to be made between the accuracy of the step length (and thus the 6MWD) and the ease to determine the step length. Another alternative is that the healthcare professional determines the patient’s step length once manually and, from then on, this manually measured step length is used in the 6MWT application. As such, patients do not have to measure their step length themselves, but the application still uses an accurate measurement of the patient’s step length. However, when this scenario is not possible, the fallback option can still be to use a formula (such as Equation (1)) to estimate the patient’s step length.

Lastly, we should also extend the 6MWT application to support patients in interpreting the results of their 6MWT. As indicated by the results of the questionnaire, some patients are unsure about what the result of their 6MWT means. The 6MWT application should include features that enable patients to interpret their results and follow up on their progress. We also assessed patients’ opinion about possible features for a 6MWT application. The results indicated that patients prefer to have features that provide feedback during the 6MWT, enable progress follow-up after the 6MWT and depict walking speed details after the 6MWT. We suggest that these features are included in the 6MWT application. The 6MWT is also used as an important assessment tool in other (chronic) diseases, e.g., neurological diseases. In this context, our elaborated 6MWT application could be interesting as well.

## 5. Conclusions

In this study, we investigated different algorithms, positions to wear a smartphone, and walking trajectories for a smartphone-measured 6MWT. For both the smartphone-measured, standardized and random walk 6MWT, the algorithm *Google Fit Processed* with the smartphone worn in a strap around the patient’s *arm* performed the best when considering both accuracy and reliability. The smartphone-measured, standardized 6MWT and the smartphone-measured, random walk 6MWT were comparable in terms of accuracy and reliability, highlighting the possibility to ease the protocol for a self-administered, smartphone-measured 6MWT in a home setting. The results of our questionnaire investigating cardiovascular disease patients’ opinion about performing a 6MWT with a smartphone application are in line with our analysis results, i.e., a preference to wear the smartphone in a strap around the arm and walk freely during the 6MWT (random walk). In summary, we can conclude that a smartphone-measured, random walk 6MWT could be a valid alternative for a standardized 6MWT supervised by a healthcare professional in a hospital setting to follow up on a patient’s functional capacity.

## Figures and Tables

**Figure 1 sensors-22-04277-f001:**
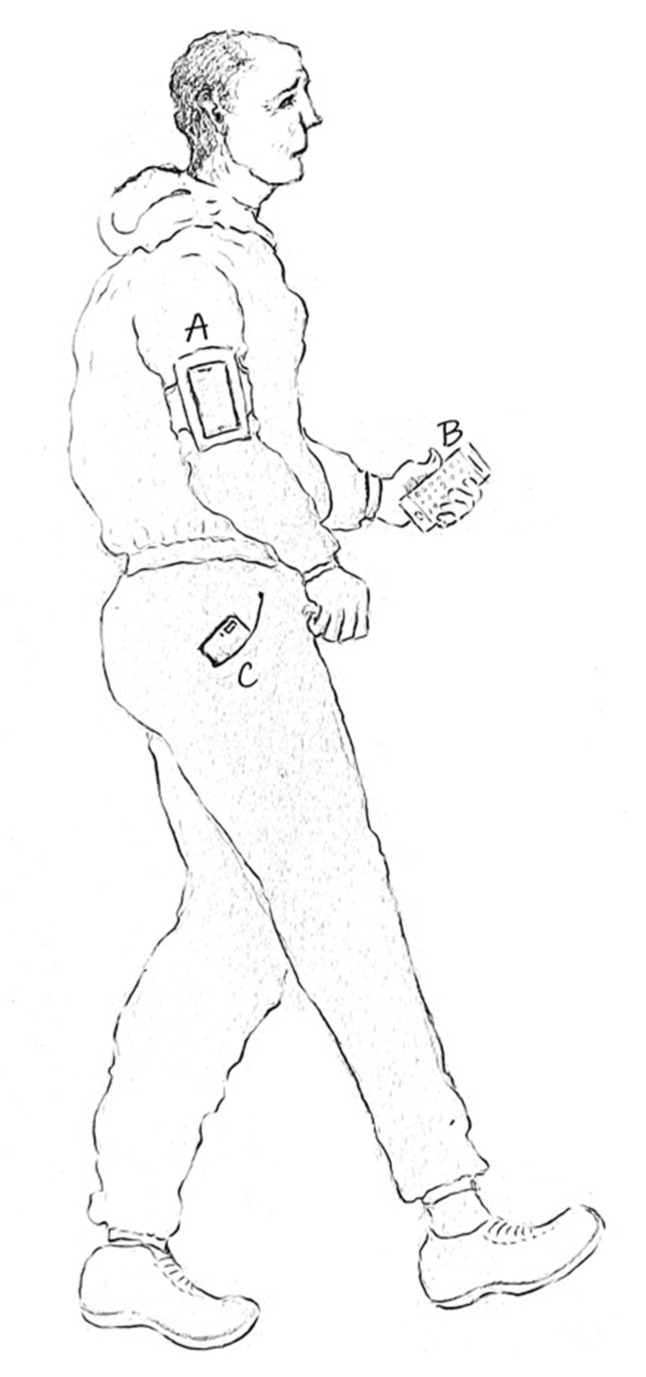
The smartphones’ positions during the six-minute walk test. (**A**)—Arm position, (**B**)—Hand position. (**C**)—Pocket position.

**Figure 2 sensors-22-04277-f002:**
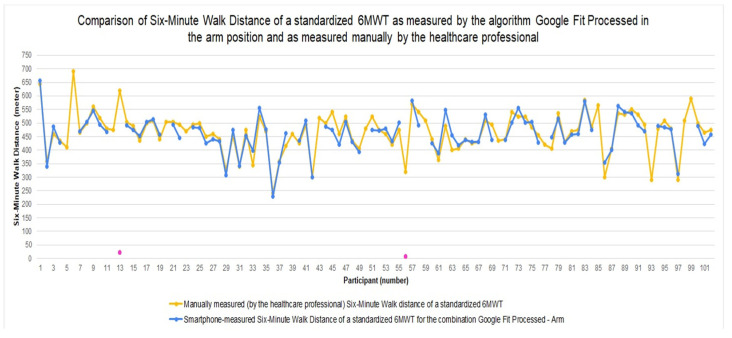
Six-minute walk distance for the standardized 6MWT as measured manually by the healthcare professional, and as measured by the algorithm *Google Fit Processed* with the smartphone in the *arm* position.

**Figure 3 sensors-22-04277-f003:**
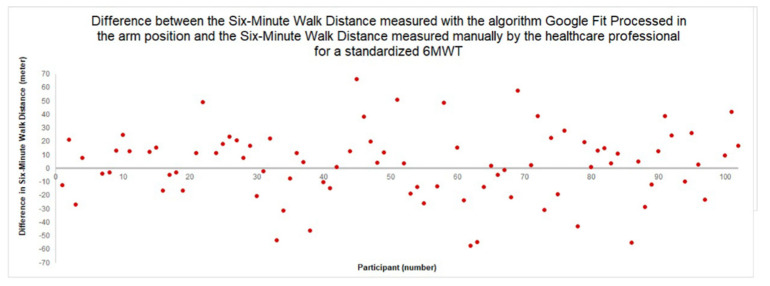
Bland–Altman plot of the difference between the 6MWD for the standardized 6MWT as measured by the algorithm *Google Fit Processed* in the *arm* position, and as measured by the healthcare professional.

**Figure 4 sensors-22-04277-f004:**
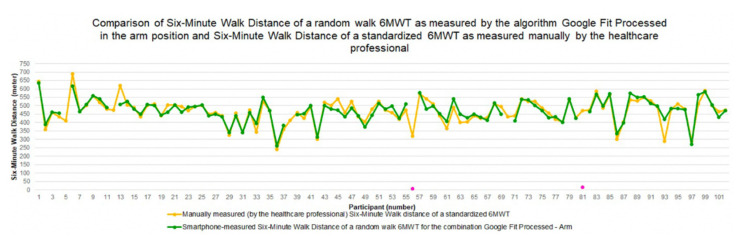
Six-minute walk distance for the random walk 6MWT as measured by the algorithm *Google Fit Processed* with the smartphone in the *arm* position and for the standardized 6MWT as measured manually by the healthcare professional.

**Figure 5 sensors-22-04277-f005:**
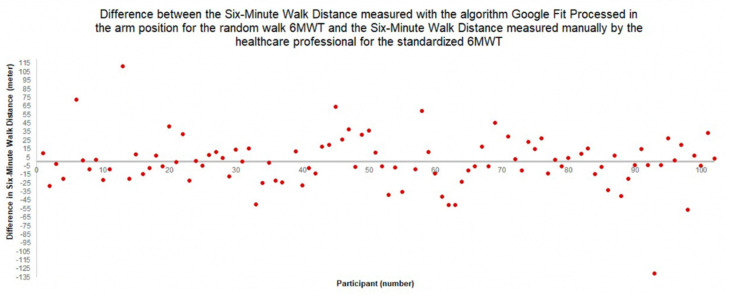
Bland–Altman plot of the difference between the six-minute walk distance for the random walk 6MWT as measured by the algorithm *Google Fit Processed* in the *arm* position and for the standardized 6MWT as measured manually by the healthcare professional.

**Figure 6 sensors-22-04277-f006:**
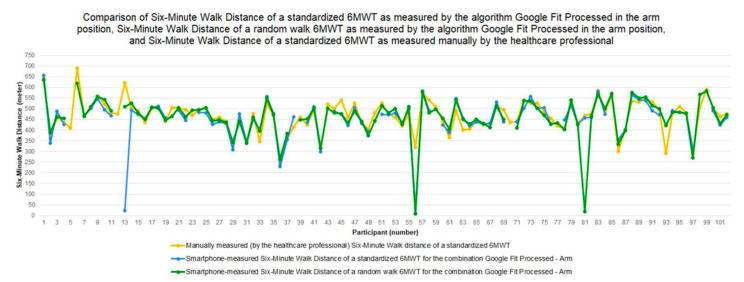
Six-minute walk distance for both the standardized 6MWT and the random walk 6MWT as measured by the algorithm *Google Fit Processed* with the smartphone in the arm position and for the standardized 6MWT as measured manually by the healthcare professional.

**Figure 7 sensors-22-04277-f007:**
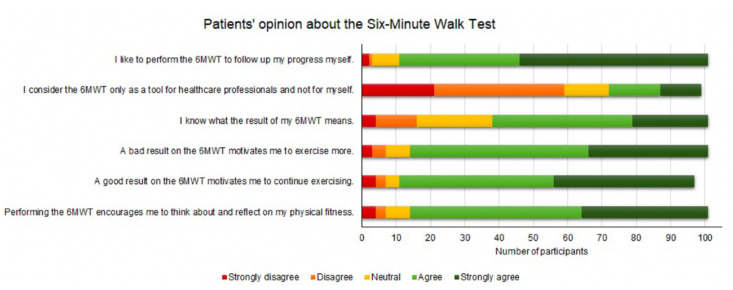
Cardiovascular disease patients’ opinion about the six-minute walk test (6MWT).

**Figure 8 sensors-22-04277-f008:**
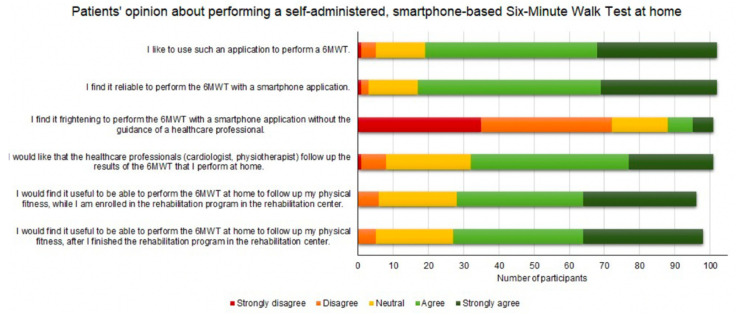
Cardiovascular disease patients’ opinion about performing a self-administered, smartphone-based six-minute walk test (6MWT).

**Figure 9 sensors-22-04277-f009:**
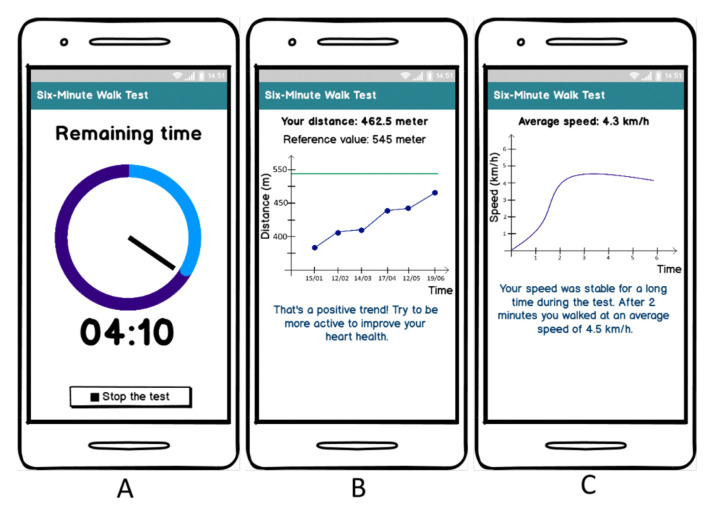
Mockups of a smartphone application to support cardiovascular disease patients in performing a self-administered six-minute walk test at home. (**A**) Feedback during the test, (**B**) progress follow-up after the 6MWT, and (**C**) walking speed details after the 6MWT. 6MWT: six-minute walk test.

**Figure 10 sensors-22-04277-f010:**
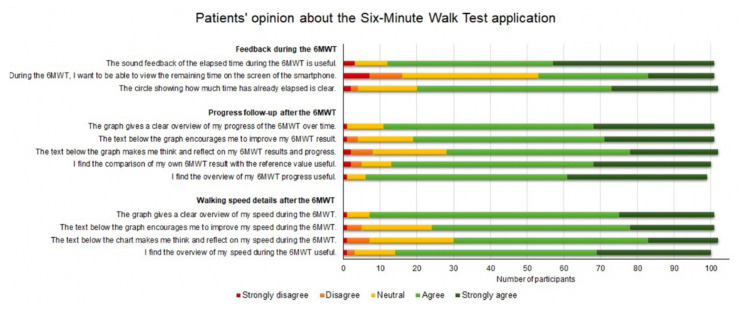
Cardiovascular disease patients’ opinion about the mockups of a six-minute walk test application. 6MWT: six-minute walk test.

**Table 1 sensors-22-04277-t001:** Results of the smartphone-measured 6MWDs of a standardized 6MWT using different algorithms and different smartphone positions. MAE: mean absolute error, SD: standard deviation.

Personalized-Threshold, Outlier-Filtered Data Set					
Algorithm	Position	MAE (m)	Median (m)	SD (m)	Error Rate
*Google Fit Raw*	Pocket	33.81	14.30	27.63	0.07
	Hand	49.06	39.95	30.90	0.11
	Arm	43.15	38.46	24.22	0.09
*Google Fit Processed*	Pocket	24.51	3.54	23.55	0.05
	Hand	27.35	7.81	24.01	0.06
	Arm	20.16	4.03	15.85	0.04
*GPS*	Pocket	43.55	−7.49	31.48	0.10
	Hand	35.06	−31.41	23.71	0.08
	Arm	35.54	−24.22	27.23	0.08
*GPS and Google Fit*	Pocket	55.17	32.88	50.47	0.11
	Hand	33.94	−8.08	28.29	0.07
	Arm	41.48	−2.62	40.73	0.09

**Table 2 sensors-22-04277-t002:** Number of missing values and number of outliers for each combination of algorithm–smartphone position for the smartphone-measured, standardized six-minute walk test.

Algorithm	Position	Number of Outliers with Personalized-Threshold, Outlier Filtering Method
*Google Fit Raw*	Pocket	0
	Hand	0
	Arm	0
*Google Fit Processed*	Pocket	0
	Hand	1
	Arm	2
*GPS*	Pocket	0
	Hand	0
	Arm	3
*GPS and Google Fit*	Pocket	36
	Hand	1
	Arm	3

**Table 3 sensors-22-04277-t003:** Results of the smartphone-measured 6MWDs of a random walk 6MWT using different algorithms and different smartphone positions. MAE: mean absolute error, SD: standard deviation.

Personalized-Threshold, Outlier-Filtered Data Set					
Algorithm	Position	MAE (m)	Median (m)	SD (m)	Error Rate
*Google Fit Raw*	Pocket Hand	36.18 53.45	22.59 48.46	28.87 38.46	0.07 0.11
	Arm	45.03	37.25	28.51	0.09
*Google Fit Processed*	Pocket Hand	24.33 31.27	−3.88 8.36	25.04 29.90	0.06 0.07
	Arm	20.56	−2.03	21.67	0.05
*GPS*	Pocket Hand	30.46 38.27	−26.62 −30.59	20.57 23.96	0.07 0.08
	Arm	36.45	−32.05	26.42	0.08
*GPS and Google Fit*	Pocket Hand	27.25 36.12	−12.78 −26.66	22.18 26.23	0.06 0.08
	Arm	31.89	26.78	24.87	0.07

**Table 4 sensors-22-04277-t004:** Number of missing values and number of outliers for each combination of algorithm–smartphone position for the smartphone-measured, random walk six-minute walk test.

Algorithm	Position	Number of Outliers with Personalized-Threshold, Outlier Filtering Method
*Google Fit Raw*	Pocket	0
	Hand	0
	Arm	0
*Google Fit Processed*	Pocket	0
	Hand	1
	Arm	2
*GPS*	Pocket Hand	0
		1
	Arm	0
*GPS and Google Fit*	Pocket	33
	Hand	1
	Arm	0

## Data Availability

Data available on request due to restrictions eg privacy or ethical.

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
