# Peer review of "A Mobile Application to Perform the Six-Minute Walk Test (6MWT) at Home: A Random Walk in the Park Is as Accurate as a Standardized 6MWT"

_sensors, 2022, doi:10.3390/s22114277_

Round 1

Reviewer 1 Report

  1. As the authors stated in "Section 4.2 Limitations", it is main weakness of the paper that the ground truth  of the random walk distance is not given. Instead, the distance is estimated from 6MWD of the healthcare professional. However, the accuracy of that distance is not verified. Thus it would be desirable if the accuracy is evaluated: by asking the healthcare professional to walk a certain random route whose total length is measured (for example, by using rolling measure).
  2. There are many smartphone-based personal (pedestrian) navigation algorithms, which could be used to estimate 6MWD more accurately. I recommend that the authors provide some comments about the possbilites.
  3. The combination of GPS and step count based estimation should give the best results. In personal navigation algorithms, there are many such algorithms. It would be desirable if authors try algorithms more optimally combining GPS and step distance.  In my personal opinion, Algorithm IV is not a good algorithm.

Author Response

Dear Editor,

Thank you for giving us the opportunity to submit a revised draft of our manuscript entitled " A Mobile Application to Perform the Six-Minute Walk Test (6MWT) at Home: A Random Walk in the Park is as Accurate as a Standardized 6MWT” to Sensors. We appreciate the time and effort that you and the reviewers have dedicated providing valuable feedback on our manuscript. We are grateful to the reviewers for their insightful comments on our paper. Here is a point-by-point response to the reviewer’s comments and concerns.

Comments from Reviewer 1

Comment:

As the authors stated in "Section 4.2 Limitations", it is main weakness of the paper that the ground truth  of the random walk distance is not given. Instead, the distance is estimated from 6MWD of the healthcare professional. However, the accuracy of that distance is not verified. Thus it would be desirable if the accuracy is evaluated: by asking the healthcare professional to walk a certain random route whose total length is measured (for example, by using rolling measure).

Response:

Thanks for this comment. The distance measured by a healthcare professionals is how 6MWT is always performed in a clinical setting. The measurement was performed following the existing guidelines. The health professional measured 6MWT is considered the gold standard. Therefore, it can be considered as a valid measuring method.

Comment:

There are many smartphone-based personal (pedestrian) navigation algorithms, which could be used to estimate 6MWD more accurately. I recommend that the authors provide some comments about the possbilites.

Response:

Thanks for this excellent comment. We have tried to describe the algorithms used in the study of Salvi et al. In the other studies using smartphone-based 6MWT, there is not much information about the algorithms used.

Comment:

The combination of GPS and step count based estimation should give the best results. In personal navigation algorithms, there are many such algorithms. It would be desirable if authors try algorithms more optimally combining GPS and step distance.  In my personal opinion, Algorithm IV is not a good algorithm.

Response:

Thank you for this comment. Algorithm IV is used in most smartphone-based 6MWT studies. Therefore, we considered it as a good option in this testing. We agree that the combination of GPS and step count seems the best option however the current study demonstrates that the Google Processed data is even more accurate.  

Reviewer 2 Report

Six-minute walk test is widely used to assess the functional exercise capacity, and this study developed a mobile application to perform 6MWT in a home setting. Different algorithms and positions were compared in this study in order to identify the optimal circumstances to perform a smartphone-based 6MWT. The topic is relevant to the scope of the journal. However, numerous outstanding issues (poor study design, poor presentation, and interpretation of the results) have drastically lowered the quality of this study. For more details, see my comments below:

Major comments

  1. The current gap in smartphone-based 6MWT should be discussed in the introduction. Have any previous studies analyzed the effect of algorithm and position on the accuracy of 6MWT? Such discussion could provide a better idea of the novelty of this study.
  2. The structure of the manuscript should be improved in the Materials and Methods section, for example, 2.1 introduced the 6MWT application and algorithms, 2.2 introduced algorithms again, and 2.3 was also related to the 6MWT application. This section should be organized properly to avoid confusion.
  3. More details should be provided for the data used in this study, it is unclear what is the difference between raw data and processed data? Is raw data a time-series data? What is the sample rate of the raw data? How the step length was determined prior to the start of 6MWT? What is the meaning of “data type” in 2.2? How the step count was estimated using the raw Google fit data?
  4. 103 patients were recruited and only 1 dropped out, why there are only 100 participants mentioned in the study design, what are the exclusion criteria for the other two patients?
  5. The MAE is over 30m for the full data set, indicating that these methods are not very accurate, even the fixed-threshold, outlier-filtered data set has a lower MAE in some methods, but only the Pocket-Google Fit Raw showed an MAE<5m, it is unclear how the step count was estimated from the raw data, hence, the reliability of this method is questionable.
  6. In Table 1, the Pocket-Google Fit Raw has an MAE of 3.81m but a median of 14.3m, such results seem unreliable.
  7. The median error might be calculated as the median of the absolute difference between 6MWDs measured in different ways.
  8. The reliability of these methods should be validated by comparing the difference in repeated measures for the same participants at different time points.
  9. The results section should be re-organized to make it more concise, some comparisons could be combined into 1 figure with 3 subfigures (i.e., figures 2, 4, and 5). The results in Table 2 and 3 have already been presented in Table 1, there is no need to list them in a separate table. Similar issues could be found in the results of the random walk test.
  10. The reviewer feels that the authors exaggerated their findings, the smartphone-measured, walk 6MWT might not be valid in a home setting due to the large MAE.

Minor comments

  1. The rationale for designing such an application for cardiac patients could be further enhanced. As there are plenty of clinical tests which were also frequently used and easy to perform, such as TUG, 10-meter walk test, Berg balance Scale test, posture sway test, et al. Why only the 6MWT was chosen for the cardiac patients in this study? Is there evidence from clinical trials supports the statement that 6MWT could be used to evaluate diagnostic and therapeutic strategies?  
  2. Please try to be more concise, and avoid the writings such as “In this section, we will xxx”, and “in this section, we briefly describe xxx”. Such information has been already provided by the title/subtitle, so it is unnecessary to repeat this.
  3. 2, Please provide more details for the four different algorithms, for example, what software was used to program these algorithms?
  4. 2, Please provide more details for the raw data and processed data
  5. Please try to use pictures such as flowcharts to demonstrate the study procedure, thus the readers could understand the study design easily.
  6. 4.2. I doubt the design that “if patients did not have a front pocket in their pants, the smartphone was put in a belt that was worn around the waist.“ as the front pocket is closer to the thigh segment, whose motion range is larger than the waist.
  7. How these 3 phones were triggered simultaneously? Please be more clear.
  8. How the error rate was calculated? How do you determine the “incorrectly measured 6MWD”?
  9. How the 6MWD was measured for random walk 6MWT by the healthcare professional?
  10. For the personalized threshold (50% of the measured 6MWD), this part is hard to understand. Have the patients already conducted 6MWT before the experiment? So the pre-measured 6MWD could be used for threshold calculation?
  11. Lien 273, MCID was suddenly mentioned here, it should be defined in the Methods Section at first.
  12. Line 278-280, “the changes in 6MWD exceed 30.5m could be considered clinically meaningful” cannot prove that MAE<30m is acceptable.
  13. Table 3, How could the SD be negative for GPS?
  14. Table 4, there are too much missing data and outliers, >10% for many cases, which should be one of the limitations of this study.

Author Response

Dear Editor,

Thank you for giving us the opportunity to submit a revised draft of our manuscript entitled " A Mobile Application to Perform the Six-Minute Walk Test (6MWT) at Home: A Random Walk in the Park is as Accurate as a Standardized 6MWT” to Sensors. We appreciate the time and effort that you and the reviewers have dedicated providing valuable feedback on our manuscript. We are grateful to the reviewers for their insightful comments on our paper. Here is a point-by-point response to the reviewer’s comments and concerns.

Comments from Reviewer 2

Comment:

The current gap in smartphone-based 6MWT should be discussed in the introduction. Have any previous studies analyzed the effect of algorithm and position on the accuracy of 6MWT? Such discussion could provide a better idea of the novelty of this study.

Response:

Thank you for this comment. We have added some additional information about the novelty of this study:

The novelty of this study is threefold: 1) It is one of the first studies to look into the precision of smartphone-based random walk 6MWT. 2) It is one of the first studies to look into the impact of various smartphone positions on the accuracy of smartphone-based 6MWT. 3) It is also one of the first studies to look into a smartphone-based 6MWT in a group of cardiac rehabilitation patients.

Comment:

The structure of the manuscript should be improved in the Materials and Methods section, for example, 2.1 introduced the 6MWT application and algorithms, 2.2 introduced algorithms again, and 2.3 was also related to the 6MWT application. This section should be organized properly to avoid confusion.

Response:

We agree with this comment. We have changed the structure and shortened the Material and Methods section.

We have changed to only section to 2.1 Algorithm and 2.2 6MWT application.

Comment:

More details should be provided for the data used in this study, it is unclear what is the difference between raw data and processed data? Is raw data a time-series data? What is the sample rate of the raw data? How the step length was determined prior to the start of 6MWT? What is the meaning of “data type” in 2.2? How the step count was estimated using the raw Google fit data?

Response:

The measurement of the step length is explained in line 134-140. The difference between the raw data and the processed data is the fact that in the processed data, the data is processed by an algorithm of Google to make the estimation more correctly. More information about the algorithms can be found in the reference.

We have tried to make it more clear between line 84-90.

Comment:

103 patients were recruited and only 1 dropped out, why there are only 100 participants mentioned in the study design, what are the exclusion criteria for the other two patients?

Response:

Thanks for pointing this out. There were 103 patients recruited and only 1 patient dropped out. So, the correct number is 102 patients. We have changed this in the manuscript.

Comment:

The MAE is over 30m for the full data set, indicating that these methods are not very accurate, even the fixed-threshold, outlier-filtered data set has a lower MAE in some methods, but only the Pocket-Google Fit Raw showed an MAE<5m, it is unclear how the step count was estimated from the raw data, hence, the reliability of this method is questionable.

Response:

In reality, we use only the personalised threshold dataset. The MAE for the best algorithm is 20m for both the standard as the random walk test. This is comparable with earlier studies testing Smartphone 6MWT applications.

Comment:

In Table 1, the Pocket-Google Fit Raw has an MAE of 3.81m but a median of 14.3m, such results seem unreliable.

Response:

Thank you for pointing this out. This is indeed wrong. We have changed the tables.

Comment:

The reliability of these methods should be validated by comparing the difference in repeated measures for the same participants at different time points.

Response:

We agree with this comment. We are planning to perform this in the future.

Comment:

The results section should be re-organized to make it more concise, some comparisons could be combined into 1 figure with 3 subfigures (i.e., figures 2, 4, and 5). The results in Table 2 and 3 have already been presented in Table 1, there is no need to list them in a separate table. Similar issues could be found in the results of the random walk test.

Response:

We have changed the results section to make it more concise. We have deleted some of the figures and tables to reduce repetitions.

Comment:

The reviewer feels that the authors exaggerated their findings, the smartphone-measured, walk 6MWT might not be valid in a home setting due to the large MAE.

Response:

We have rewritten our conclusions. However, we still believe that our smartphone-measured random walk 6MWT is a valid option if you compare it with earlier studies.

Comment:

The rationale for designing such an application for cardiac patients could be further enhanced. As there are plenty of clinical tests which were also frequently used and easy to perform, such as TUG, 10-meter walk test, Berg balance Scale test, posture sway test, et al. Why only the 6MWT was chosen for the cardiac patients in this study? Is there evidence from clinical trials supports the statement that 6MWT could be used to evaluate diagnostic and therapeutic strategies?  

Response:

The tests that you mentioned are important tests in a geriatric assessment for frailty. It is true that frailty is common in cardiovascular patients. However, the 6MWT is a well-known and often used test to follow-up the functional capacity of cardiovascular patients. We have added a statement that 6MWT could be used to evaluate diagnostic and therapeutic strategies in line 48-57.

Comment:

Please try to be more concise, and avoid the writings such as “In this section, we will xxx”, and “in this section, we briefly describe xxx”. Such information has been already provided by the title/subtitle, so it is unnecessary to repeat this.

Response:

Thank you for pointing this out. We have tried to make it more concise. We have left out the writings above.

Comment:

Please provide more details for the four different algorithms, for example, what software was used to program these algorithms?

Response:

We have added additional information about the four algorithms in chapter 2.1. We have added the programming languages that were used. More information about the algorithms could also be found in the following references.

Comment:

Please provide more details for the raw data and processed data

Response:

The difference between the raw data and the processed data is the fact that in the processed data, the data is processed by an algorithm of Google to make the estimation more correctly. More information about the algorithms can be found in the reference.

We have tried to make it more clear between line 84-90.

Comment:

I doubt the design that “if patients did not have a front pocket in their pants, the smartphone was put in a belt that was worn around the waist.“ as the front pocket is closer to the thigh segment, whose motion range is larger than the waist.

Response:

Thanks for this comment. We agree that it is possible that there could occur differences due to the placement of the smartphone.

Comment:

How these 3 phones were triggered simultaneously? Please be more clear.

Response:

We have added information on this on line 164.

The healthcare professional had a master smartphone that could trigger the start of the 6MWT on the different smartphones simultaneously via Bluetooth. As such, the healthcare professional could trigger the start of the 6MWT after all smartphones were correctly positioned on the patient's body.

Comment:

How the error rate was calculated? How do you determine the “incorrectly measured 6MWD”?

Response:

The error rate was calculated following this formula:

The mean of all Percent error outcomes which was determined by: the absolute value of the difference of the measured value and the actual value divided by the actual value and multiplied by 100.

Comment:

How the 6MWD was measured for random walk 6MWT by the healthcare professional?

Response:

This was not measured. It was compared with the distance measured in the standard 6MWT by the health professional. The goal of the random walk 6MWT is to correlated as good as possible with the standard 6MWT distance.

Comment:

For the personalized threshold (50% of the measured 6MWD), this part is hard to understand. Have the patients already conducted 6MWT before the experiment? So the pre-measured 6MWD could be used for threshold calculation?

Response:

The participants did not conduct a 6MWT before the experiment. The first test in which the patient performed a standardized 6MWT was used for the threshold calculation.

There were 2 tests in this study.

1) Standardized fixed path 6MWT measured by the health professional and at the same time by the 3 smartphones

2) A random walk 6MWT measured by the three smartphones

Comment:

Lien 273, MCID was suddenly mentioned here, it should be defined in the Methods Section at first.

Response:

Thanks for pointing this out. We have added additional information about MCID in the methods section.

Comment:

Table 3, How could the SD be negative for GPS?

Response:

Thank you for pointing this out. This is indeed wrong. We have changed the tables.

Sincerely,

Dr. Martijn Scherrenberg, on behalf of the authors

Round 2

Reviewer 1 Report

There are no further comments.

Author Response

Dear Editor,

Thank you for giving us the opportunity to submit a revised draft of our manuscript entitled " A Mobile Application to Perform the Six-Minute Walk Test (6MWT) at Home: A Random Walk in the Park is as Accurate as a Standardized 6MWT” to Sensors. We appreciate the time and effort that you and the reviewers have dedicated providing valuable feedback on our manuscript. We are grateful to the reviewers for their insightful comments on our paper. Here is a point-by-point response to the reviewer’s comments and concerns.

Comments from Reviewer 1

No comments

Sincerely,

Dr. Martijn Scherrenberg, on behalf of the authors

Reviewer 2 Report

The authors have addressed most of my concerns except those below:

1. Regarding my concern on the MAE over 30m in this study, the authors argued that 30m is good, as the MAE for the best algorithm is just 20m, but such information is not provided in the manuscript. The authors should try to convince the readers that such MAE is a valid option, but not just ask the readers to do the comparison with earlier studies. 

2. Regarding the study design "if patients did not have a front pocket in their pants, the smartphone was put in a belt that was worn around the waist".  The authors agreed that it is possible that there could occur differences due to the placement of the smartphone. Therefore, this is a limitation of this study, while, the authors did not add this information in the Discussion section. 

3. The authors did not explain what is "data type" in 2.1

Author Response

Dear Editor,

Thank you for giving us the opportunity to submit a revised draft of our manuscript entitled " A Mobile Application to Perform the Six-Minute Walk Test (6MWT) at Home: A Random Walk in the Park is as Accurate as a Standardized 6MWT” to Sensors. We appreciate the time and effort that you and the reviewers have dedicated providing valuable feedback on our manuscript. We are grateful to the reviewers for their insightful comments on our paper. Here is a point-by-point response to the reviewer’s comments and concerns.

Comments from Reviewer 2

Comment:

Regarding my concern on the MAE over 30m in this study, the authors argued that 30m is good, as the MAE for the best algorithm is just 20m, but such information is not provided in the manuscript. The authors should try to convince the readers that such MAE is a valid option, but not just ask the readers to do the comparison with earlier studies. 

Response:

Thanks for pointing this out. We have elaborated on this in the discussion. We have added this in line 613-616.

Comment:

Regarding the study design "if patients did not have a front pocket in their pants, the smartphone was put in a belt that was worn around the waist".  The authors agreed that it is possible that there could occur differences due to the placement of the smartphone. Therefore, this is a limitation of this study, while, the authors did not add this information in the Discussion section. 

Response:

Thanks for pointing this out. We have added this to the limitations in line 665-667.

Comment:

The authors did not explain what is "data type" in 2.1

Response:

We have added an explanation of “data type” and “data source” in line 86-92.

Sincerely,

Dr. Martijn Scherrenberg, on behalf of the authors